# Multifunctional Nanoemulsified *Clinacanthus nutans* Extract: Synergistic Anti-Pathogenic, Anti-Biofilm, Anti-Inflammatory, and Metabolic Modulation Effects against Periodontitis

**DOI:** 10.3390/biology13100815

**Published:** 2024-10-11

**Authors:** Sirintip Pechroj, Thida Kaewkod, Pachara Sattayawat, Angkhana Inta, Sureeporn Suriyaprom, Teerapong Yata, Yingmanee Tragoolpua, Itthayakorn Promputtha

**Affiliations:** 1Department of Biology, Faculty of Science, Chiang Mai University, Chiang Mai 50200, Thailand; sirintip253771@gmail.com (S.P.); thida.kaewkod@cmu.ac.th (T.K.); pachara.sattayawat@cmu.ac.th (P.S.); angkhana.inta@cmu.ac.th (A.I.); sureeporn.suriyaprom@cmu.ac.th (S.S.); yingmanee.t@cmu.ac.th (Y.T.); 2Multidisciplinary and Interdisciplinary School, Chiang Mai University, Chiang Mai 50200, Thailand; 3Office of Research Administration, Chiang Mai University, Chiang Mai 50200, Thailand; 4Premier Innova Co., Ltd., Nong Bon, Prawet, Bangkok 10250, Thailand; teerapong.y@pinno.premier.co.th

**Keywords:** anti-oral plaque, diabetic periodontal disease, gum healing, multifunctional extract, nanoemulsion delivery, oral pathogenic bacteria

## Abstract

**Simple Summary:**

This study investigates the health benefits of *Clinacanthus nutans* extracts, specifically the 95% ethanol extract and its nanoemulsified form, highlighting their potential for oral health and diabetes management. These extracts exhibit strong antibacterial activity against harmful oral bacteria, including *Streptococcus mutans* and *Staphylococcus aureus*, which are important for preventing periodontal diseases. They also possess significant anti-biofilm properties, which help reduce dental plaque formation. In addition, the extracts inhibit the enzyme α-glucosidase, suggesting they may aid in regulating blood sugar levels in diabetes. Their anti-inflammatory effects, demonstrated by reduced nitric oxide production, indicate potential benefits for treating oral infections and inflammation. Nanoemulsification enhances the extracts’ solubility, stability, and bioavailability, improving their therapeutic effectiveness. Overall, this study suggests that *Clinacanthus nutans* extracts, particularly in nanoemulsified form, could be developed into new treatments for oral health issues and metabolic disorders such as diabetes. Further research is needed to confirm their safety and effectiveness in clinical applications.

**Abstract:**

This study investigates the therapeutic potential of *Clinacanthus nutans* extracts, focusing on the 95% ethanol (95E) extract and its nanoemulsified form, against oral pathogens and their bioactive effects. The findings demonstrate potent antibacterial activity against *Streptococcus mutans* and *Staphylococcus aureus*, essential for combating periodontal diseases, and significant anti-biofilm properties crucial for plaque management. Additionally, the extracts exhibit promising inhibitory effects on α-glucosidase enzymes, indicating potential for diabetes management through glucose metabolism regulation. Their anti-inflammatory properties, evidenced by reduced nitric oxide production, underscore their potential for treating oral infections and inflammation. Notably, the nanoemulsified 95E extract shows higher efficiency than the conventional extract, suggesting a multifunctional treatment approach for periodontal issues and metabolic disorders. These results highlight the enhanced efficacy of the nanoemulsified extract, proposing it as an effective treatment modality for periodontal disease in diabetic patients. This research offers valuable insights into the development of innovative drug delivery systems using natural remedies for improved periodontal care in diabetic populations.

## 1. Introduction

Periodontal disease, a chronic inflammatory condition affecting the supporting structures of the teeth [1], poses a significant challenge, especially in diabetic patients who exhibit compromised healing responses and heightened susceptibility to infections. Periodontitis may impair insulin resistance and glycemic control [2]. The mechanisms underlying these associations are complex and multifactorial, involving interactions between oral bacteria, inflammatory mediators, and systemic inflammation. Major pathogens within the oral cavity, particularly bacteria triggering biofilm accumulation and chronic inflammation, include *Streptococcus mutans*, *Streptococcus pyogenes*, *Klebsiella pneumoniae*, *Porphyromonas gingivalis*, *Prevotella intermedia*, *Staphylococcus aureus*, *Tannerella forsythia*, and *Treponema denticola* [3]. Bacteria induce periodontitis by forming biofilms. They interact with the host’s immune system and release toxins as well as enzymes that trigger inflammation and tissue destruction. As the biofilm matures, it becomes increasingly resistant to antimicrobial agents and immune defenses, allowing bacteria to persist and exacerbate periodontal disease [4]. Diabetic mellitus is known to promote the expression of pro-inflammatory cytokines and mediators in human periodontal tissues. Patients with diabetes and periodontal disease exhibit higher levels of inducible nitric oxide synthase (iNOS), leading to excessive nitric oxide (NO) production [5]. In such hyperglycemic conditions, the activity of α-glucosidase becomes notably pronounced [6]. Therapeutic interventions for periodontitis focus on reducing infectious and inflammatory challenges, often involving the removal of pathogenic biofilms and suppression of inflammation. 

*Clinacanthus nutans* (Burm.f.) Lindau, a medicinal plant rich in bioactive compounds, exhibits diverse pharmacological properties, including analgesic, anti-inflammatory, antiviral, antibacterial, anti-biofilm, antioxidant, protect against free radical-induced hemolysis, immunomodulatory, and antidiabetic properties [7,8,9,10,11,12,13]. This plant is listed in Thai herbal pharmacopeias with reported uses of crushed leaves soaked in 40% ethanol as anti-viral and anti-inflammatory agents. The dried leaf powder of *C. nutans* contains various phytochemical constituents, including phenolic compounds, flavonoids, alkaloids, saponins, triterpenoids, diterpenes, steroids, phytosterols, tannins, carbohydrates, and proteins/amino acids [14]. In a clinical study, *C. nutans* reduced recurrent aphthous ulcer healing time [15]. This finding suggests that *C. nutans* may affect cell migration in wound healing. The extract of *C. nutans* recently experienced more thorough investigation for its potential in preventing oral diseases, particularly plaque-related diseases, such as dental caries [16]. However, the restricted dosage of drugs has impeded their widespread utilization. Consequently, developing nanoparticles containing *C. nutans* extract offers a promising strategy to address periodontal disease complexities, particularly in diabetic patients. 

Nanoemulsions are increasingly recognized as highly effective systems for drug delivery due to their nanoscale droplets, which offer a significantly larger surface area for enhanced drug absorption. Typically composed of oil, water, and emulsifiers or their blends, nanoemulsions exhibit varying appearances from transparent to translucent, with internal droplets typically ranging from 20 to 200 nm in size. This size range ensures slow destabilization kinetics, thereby enhancing their stability. Production methods for nanoemulsions primarily involve high-energy and low-energy approaches. High-energy methods utilize equipment like high-pressure homogenizers, microfluidizers, or ultrasonicators to achieve fine emulsification. In contrast, low-energy methods facilitate nanoemulsion formation using simpler equipment, with both methods capable of scaling up for large-scale production.

The research gap identified in the provided article highlights the need for innovative therapeutic strategies to manage periodontal disease in diabetic patients, who exhibit compromised healing and heightened susceptibility to infections. Current treatments struggle with the complexities of disease mechanisms, particularly the resistance of bacterial biofilms to antimicrobial agents and immune defenses. Although *C. nutans* shows promise due to its anti-inflammatory, anti-biofilm, antimicrobial, and anti-α-glucosidase properties, its limited dosage has hindered widespread use. The development of nanoparticles containing *C. nutans* extract could enhance targeted delivery and bioavailability, addressing the intricate interactions between oral bacteria, inflammatory mediators, and systemic inflammation. Additionally, the enzyme α-glucosidase, which plays a crucial role in microbial growth and becomes notably active under hyperglycemic conditions, is a key target for mitigating microbial proliferation and inflammation in periodontal disease. However, the integration of natural remedies with nanotechnology for periodontal disease management remains insufficiently explored, necessitating further research to validate its efficacy and potential benefits. The synergistic nanoparticle strategy aims to inhibit periodontal pathogens and modulate inflammatory gene expression, thereby offering a comprehensive approach to managing periodontal diseases. By encapsulating the bioactive compounds of *C. nutans* within nanoparticles, it is possible to achieve targeted delivery, sustained release, and enhanced bioavailability. This approach not only improves the therapeutic efficacy of *C. nutans* extract but also minimizes potential side effects. This study aims to develop and evaluate a nanoemulsion formulation of *C. nutans* extract to enhance its antibacteria, anti-biofilm, anti-α-glucosidase, and anti-inflammatory effects for the effective management of periodontal disease in diabetic patients.

## 2. Materials and Methods

### 2.1. Preparation of C. nutans Extracts

The finely ground dried leaf powder of *C. nutans* was obtained from Lampang Province (Lampang Herbs Shop, Lampang, Thailand). Powdered samples were soaked in various solvents (95% methanol, 95% ethanol, 70% ethanol at room temperature for 72 h, and water at 45 °C for 3 h) at a ratio of 1:10 (*w*/*v*) within a light-shielded container with infrequent shaking. Subsequently, the extracts were filtered through Whatman No. 1 filter paper (GE Healthcare, East Bridgewater, MA, USA). Residues from *C. nutans* were subjected to an additional extraction cycle, and all resulting extracted solutions were combined to form a uniform solution. The extract solution was evaporated using a rotary evaporator (Rotavapor^®^ R-300, BÜCHI Labortechnik AG, Flawil, Switzerland) at 45 °C to remove the solvent. The crude extracts were lyophilized by a freeze dryer (Lyovapor™ L-200 Freeze Dryer, BÜCHI Labortechnik AG, Switzerland) until dryness. The lyophilization process involves freezing the crude extract at −80 °C for 12 h, followed by a primary drying phase, where the temperature is gradually raised to −20 °C under vacuum for 24 h, allowing sublimation to occur. Finally, the secondary drying phase involves increasing the temperature to 10 °C for an additional 12 h to remove any remaining bound water. The entire freeze-drying process takes 48 h. The weight of the lyophilized extracts was measured, and the percentage yield of each extract was calculated by using the equation below. The extracts, labeled as 95M, 95E, 70E, and W according to their respective extraction solvents, were stored in a frozen state until further use. For all treatment applications, the extracts were dissolved in DMSO.
Percentage Yield = (Final weight of extract/Initial weight of extract) × 100%

### 2.2. Growth Inhibition Assay of Periodontitis Pathogenic Bacteria

#### 2.2.1. Agar-Well Diffusion Assay

Four common oral pathogenic bacteria—*Streptococcus mutans, Streptococcus pyogenes, Klebsiella pneumoniae*, and *Staphylococcus aureus*—were cultured at 37 °C for 24 h. The bacterial cultures were adjusted for turbidity to a 0.5 McFarland standard using a spectrophotometer at 600 nm (OD 0.08–0.1). The turbidity-adjusted bacterial suspensions were evenly distributed across Mueller–Hinton agar plates using sterile cotton swabs. Wells of 10 mm in diameter were created in the agar using a cork borer. All extracts, including *C. nutans* extracts, were dissolved in DMSO at a stock concentration of 500 mg/mL. The extracts were dispensed into the wells, and gentamicin served as a positive control. The inoculated plates were incubated at 37 °C for 24 h, after which the antibacterial activity was assessed by measuring the diameter of the clear zones of inhibition around each well, indicating bacterial growth inhibition. The results were compared to the inhibition zones produced by gentamicin. The experiment was replicated three times to ensure reproducibility, and the average diameters of the inhibition zones were calculated and reported with standard deviations to reflect consistency in antibacterial activity.

#### 2.2.2. MIC and MBC Assays

To evaluate the antibacterial efficacy of *C. nutans* extracts, minimal inhibitory concentration (MIC) and minimum bactericidal concentration (MBC) assays were performed on the four tested pathogenic bacteria, *S. mutans*, *S. pyogenes*, *K. pneumoniae*, and *S. aureus*. All bacteria were cultured at 37 °C for 24 h, and the cultures were adjusted to a 0.5 McFarland standard using a spectrophotometer at 600 nm (OD 0.08–0.1). The extracts were dissolved in DMSO at a stock concentration of 500 mg/mL. MIC was determined using the broth dilution method, where serial two-fold dilutions of the extracts (ranging from 0 to 500 mg/mL) were prepared in a 96-well microtiter plate. Turbidity-adjusted bacterial suspensions were added to each well at a 1:1 ratio with the extract dilutions and incubated at 37 °C for 24 h. The MIC endpoint was identified as the lowest concentration of the extract that inhibited visible bacterial growth, indicated by the absence of turbidity in the well. Following MIC determination, MBC was assessed by collecting samples from wells showing no visible bacterial growth and streaking them onto fresh Mueller–Hinton agar plates, which were incubated at 37 °C for 24 h. The MBC was defined as the lowest concentration of the extract that resulted in no bacterial growth on the agar plates. All tests were conducted in triplicate, and the MIC and MBC values were calculated as the average concentrations from the replicate assays, with standard deviations computed to represent variability. Gentamicin was used as a positive control to validate the assay results. The MIC and MBC values provided quantitative measures of the extract’s inhibitory and bactericidal properties, respectively, highlighting the potential therapeutic applications of *C. nutans* extracts for treating infections caused by the tested oral pathogens.

### 2.3. Evaluation of Anti-Biofilm Formation and Biofilm Degradation Efficiency

To assess the anti-biofilm formation and biofilm degradation efficiency of *C. nutans* extracts, bacterial cultures were prepared at a concentration of 10^8^ cells/mL and inoculated into a 96-well culture plate, with certain wells treated with the extracts at different concentrations ranging from 62.5–250 mg/mL. For biofilm inhibition assessment, the plate was then incubated at 37 °C for 24 h to allow for biofilm formation. To investigate biofilm degradation, pre-formed biofilms were prepared under similar conditions for 24 h prior to extract treatment. Post-incubation, the supernatant was removed, and the wells were gently washed twice with phosphate-buffered saline (PBS) to remove non-adherent bacteria. The biofilms were then stained with 0.4% crystal violet dye for 20 min, followed by thorough washing with PBS to remove excess dye. The culture plate was air-dried, and the stained biofilms were solubilized with 95% ethanol. The optical density was measured at 592 nm using a microplate reader, with the absorbance value serving as an indicator of biofilm biomass. The obtained data were used to calculate the inhibition of biofilm formation by comparing the absorbance values of treated and untreated samples. This methodology, executed in triplicate, provides a robust approach to evaluating both the preventive and therapeutic potential of *C. nutans* extracts against biofilm formation and pre-formed biofilms, supporting their potential application as anti-biofilm agents in clinical and therapeutic settings.

### 2.4. Assessment of α-Glucosidase Enzyme Inhibition

In a 96-well microplate, the assessment of α-glucosidase enzyme inhibition by *C. nutans* extracts was conducted following a standardized protocol. All extracts were dissolved in DMSO at a stock concentration of 500 mg/mL. A reaction mixture was prepared by combining 50 μL of phosphate buffer (pH 6.8), 20 μL of *C. nutans* extract at various concentrations (0–500 mg/mL), and 10 μL of α-glucosidase enzyme solution (1 U/mL). This mixture was incubated at 37 °C for 15 min to allow for enzyme and extract interaction. Subsequently, 20 μL of p-nitrophenyl-α-D-glucopyranoside (p-NPG) substrate solution (5 mM) was added to initiate the enzymatic reaction, which proceeded for an additional 20 min at 37 °C. To terminate the reaction, 50 μL of 0.1 M Na_2_CO_3_ solution was added to each well. Acarbose, a known α-glucosidase inhibitor, was included as a positive control for comparative analysis. The absorbance of the reaction mixture was measured at 405 nm using a Multiplate Reader immediately after adding Na_2_CO_3_. The percentage inhibition of α-glucosidase enzyme activity by *C. nutans* extracts was calculated. All experiments were performed in triplicate, and the results were expressed as mean percentage inhibition ± standard deviation. Statistical analysis was conducted to determine significant differences between the inhibitory effects of *C. nutans* extracts and the positive control acarbose.

### 2.5. Cytotoxicity Assay

#### 2.5.1. Cell Lines and Cell Culture

Murine macrophage cell line RAW264.7 was obtained from the American Type Culture Collection (ATCC, TIB-71TM, Rockville, MD, USA) and maintained in Dulbecco’s Modification of Eagle’s Medium (DMEM) (Gibco, Life Technologies, Grand Island, NY, USA) supplemented with 10% inactivated fetal bovine serum (FBS) and 1% penicillin–streptomycin. Cells were cultured and maintained at 37 °C in a 5% CO_2_ incubator SL (SHEL LAB, Cornelius, OR, USA). Cells were harvested within 20 passages using the cell scraper without addition of trypsin for further analysis according to the manufacturer’s instructions.

#### 2.5.2. In Vitro Assay for Cytotoxic Activity

The MTT cell viability assay was used to access the cytotoxicity of *C. nutans* extracts on RAW264.7 cells, as described previously with modification [17]. Briefly, all extracts (95M, 95E, 70E, W) were reconstituted using DMSO to 100 mg/mL and further diluted to required concentrations (1–100 μg/mL) using ultra-purified sterile water prior to the assays. Cells were treated with various concentrations of extracts or 0.1% DMSO (negative control) for 72 h before the reaction was terminated with the MTT reagent. The absorbance was recorded at a test wavelength of 570 nm and a reference wavelength of 630 nm using the Tecan Infinite F200 plate reader (Männedorf, Switzerland). The mean absorbance for the negative control (0.1% DMSO) was normalized as 100%.

### 2.6. Inhibition of Nitric Oxide (NO) Production

In the presence of lipopolysaccharide (LPS) from *Escherichia coli* O111:B4 on the RAW264.7 macrophage cells, NO is generated by inducible NO synthase in macrophages as a hallmark of inflammation [18,19]. The production of NO can be quantified by measuring the level of nitrite production, the stable metabolite of NO, as described in the Griess assay [20]. RAW264.7 macrophages were plated at 1 × 10^6^ cells/mL in a 96-well plate and treated with or without extracts (95M, 95E, 70E, W) at concentrations of 0.08–2.5 mg/mL for 24 h, followed by stimulation with or without LPS 1 µg/mL for 24 h. Supernatants (50 μL) were removed and mixed with equal amounts of Griess reagents [20]. The solutions were then left for 10 min at room temperature before measurement on a microplate reader (Dynex Technologies, Denkendorf, Germany) at 540 nm. The NO inhibition percentage was calculated, and the IC50 of NO production, indicating the concentration at which extracts inhibited 50% of LPS-induced NO production, was determined.

### 2.7. Preparation of Nanoemulsion of the 95% Ethanol Extract of C. nutans (95E)

The nanoemulsion formulation was prepared by incorporating a 95% ethanolic extract of *C. nutans* (95E) at a final concentration of 40 mg/mL, designated as the active pharmaceutical ingredient (API). This concentration was strategically selected based on prior screening results that revealed an IC50 of approximately 17.19 mg/mL, with nearly 100% inhibition observed at approximately 40 mg/mL against α-glucosidase activity. To ensure the incorporation of the API, visual inspection of the nanoemulsion was performed to confirm the absence of phase separation, complemented by preliminary bioactivity assays against a standard to validate that the desired therapeutic effects were preserved within the formulation. The preparation of the Nanostructured Lipid Carriers (NLCs) commenced with the oil phase, consisting of Medium-Chain Triglyceride (MCT) oil (20%), the 95E extract (1%), Span 80 (12%), and bee wax (2%), resulting in an estimated total lipid concentration of 500 mg/mL, contingent upon the assumption of 100% API incorporation. Next, the aqueous phase was prepared by weighing the Tween 20 (3%), glycerin (2%), poloxamer 188 (2%), and phenoxyethanol (0.1%) and adding water to 100%. This solution was agitated at a speed of 300 rpm while being heated to 75 °C, ensuring complete dissolution of all components into a uniform mixture. Subsequently, the aqueous phase was introduced into the oil phase while mechanically stirring at 300 rpm for 20 min at 75 °C to form a pre-emulsion. The nanoparticles were further reduced in size by employing a homogenizer (WIGGENS, Straubenhardt, Germany) for a duration of 10 min. This process was carried out to ensure the resulting solution’s homogeneity and achieve the desired nanoparticle size. Therefore, the size of the nanoemulsion was investigated by NanoSizer. The nanoemulsion stock solution was diluted in deionized water at a 1:10,000 (*v*/*v*) ratio. A 100 µL aliquot of the diluted solution was then transferred to a cuvette for analysis. Particle size distribution was determined using dynamic light scattering (DLS) with a Horiba SZ-100V2 instrument (Horiba, Kyoto, Japan). To ensure accuracy and reproducibility, at least three consecutive measurements were performed. The average particle diameter was calculated from the peak maxima of the DLS size distribution curves; nanoparticle size must not exceed 200 nm. Subsequently, the biological activities of the nanoemulsion were evaluated. Additionally, control lipid nanoparticles devoid of the API were synthesized to elucidate their effects, containing the same lipid components while excluding the 95E extract. Comparative assessments of the biological activities of both the API-loaded nanoemulsion and control lipid nanoparticles were undertaken to determine any baseline activity attributable to the lipid matrix alone. It is pertinent to acknowledge that the assumption of 100% API incorporation may not fully encapsulate the inherent variability in encapsulation efficiency across different batches; consequently, while preliminary testing was conducted to ascertain activity, this study’s limitations include the absence of detailed quantitative measures of incorporation efficiency.

### 2.8. Statistical Analysis

All results were expressed as means ± S.D. of at least three independent experiments. Statistical significance values were analyzed with the Tukey test for multiple comparisons following using ANOVA. 

## 3. Results

### 3.1. Percentage Yield of Crude Extracts of Clinacanthus nutans Leaves

After the evaporation of the extract’s solvent by rotary evaporator and freeze drying, the weight was measured to calculate the percentage yield of each crude extract. The percentage yields of 95M, 95E, 70E, and W extracts were 11.88%, 8.30%, 17.60%, and 19.50%, respectively.

### 3.2. Antibacterial Efficiency of C. nutans Extracts

The antibacterial efficacy of *C. nutans* extracts (95M, 95E, 70E, W) was evaluated using agar-well diffusion and MIC/MBC assays against four oral pathogenic bacteria: *S. mutans, S. pyogenes, K. pneumoniae*, and *S. aureus*. At a concentration of 500 mg/mL, the 95M, 95E, and 70E extracts demonstrated significant inhibitory activity against all tested bacteria, with no statistically significant differences observed (*p* < 0.05). Conversely, the water (W) extract showed no antibacterial activity against any of the bacterial species (Figure 1). MIC and MBC values were consistent across the methanol, ethanol (95% and 70%), and water extracts against *K. pneumoniae* and *S. pyogenes*, both measuring at 250 mg/mL. However, the W extract exhibited MIC and MBC values exceeding this threshold. For *S. aureus*, the 95E and 95M extracts displayed the lowest MIC and MBC values at 125 mg/mL, while the 70E and W extracts showed values of 250 mg/mL and greater than 250 mg/mL, respectively. Additionally, the 70E and 95M extracts demonstrated MIC and MBC values at 125 mg/mL against *S. mutans* (Table 1). These findings support previous studies and suggest that these *C. nutans* extracts hold promise for further research as a potent inhibitor of these oral bacteria.

### 3.3. Anti-Biofilm Formation and Biofilm Degradation Efficacy of C. nutans Extracts

In the anti-biofilm assays, varying concentrations of *C. nutans* extracts in different solvents were tested based on sub-MBC values against each bacterial species, as detailed in Table 2. All extracts exhibited distinct levels of biofilm inhibition and degradation across *S. mutans, S. pyogenes, K. pneumoniae*, and *S. aureus*. Overall, the 95E extract demonstrated the highest potential for inhibiting biofilm formation and degradation against *S. mutans* and *S. pyogenes*. The 95M extract also showed potential activity; however, the methanol extract is not suitable for use in the oral cavity. Therefore, the 95E extract was selected for further studies.

### 3.4. Inhibition of α-Glucosidase Enzyme Activity by C. nutans Extracts

The inhibitory effects of *C. nutans* extracts on α-glucosidase enzyme activity were evaluated, with the 70E extract demonstrating the most significant effect, as indicated by IC50 values presented in Table 3. Following this, the 95M, 95E, and W extracts showed progressively lower inhibition of enzyme activity. Notably, all extracts exhibited stronger inhibitory effects compared to acarbose, a standard enzyme inhibitor. These findings corroborate the outcomes of our study, underscoring the potential of *C. nutans* extracts as effective inhibitors of α-glucosidase enzyme activity.

### 3.5. Cytotoxicity Effect of C. nutans Extracts

Cell viability assays indicated that all extracts remained cell survival rates above 70% at a concentration of 2.5 mg/mL (Table 4), with IC50 values for cytotoxicity ranging from 3.07 to 5.51 mg/mL (Table 5).

### 3.6. Anti-Inflammatory Activity of C. nutans Extracts

The anti-inflammatory activity of *C. nutans* extracts (95M, 95E, 70E, W) was evaluated at concentrations ranging from 0.08 to 2.5 mg/mL. Subsequent investigation focused on the extracts’ ability to inhibit nitric oxide (NO) secretion in RAW264.7 macrophage cells stimulated with LPS. The results on Table 6 demonstrated concentration-dependent inhibition of NO production by the 95M and 95E extracts, both exhibiting an IC50 of 0.10 mg/mL. The 70E extract showed an IC50 of 0.15 mg/mL, while the W extract exhibited less than 50% inhibition at concentrations below 2.5 mg/mL (Table 7). These findings are consistent with previous studies highlighting the potent anti-inflammatory properties of *C. nutans* extracts, particularly those extracted with ethanol solvents, in suppressing inflammatory responses by reducing NO secretion (Table 7).

Based on the comprehensive evaluation of *C. nutans* extracts for their antibacterial, anti-biofilm, anti-α-glucosidase, and anti-inflammatory activities, the 95% ethanol (95E) extract consistently demonstrated potent biological effects across multiple assays. Specifically, the 95E extract showed significant antibacterial activity against oral pathogens (*S. mutans*, *S. pyogenes*, *K. pneumoniae*, and *S. aureus*), effective inhibition and degradation of biofilms, strong inhibition of α-glucosidase enzyme activity, and potent anti-inflammatory activity by reducing NO secretion in stimulated macrophages. These findings support the selection of the 95E extract as the most promising candidate for further development, including its use in synthesizing nanoemulsions to potentially enhance its bioavailability and therapeutic efficacy. Consequently, they were utilized to investigate the anti-inflammatory, anti-biofilm of bacteria, and anti-α-glucosidase effects by inhibiting the α-glucosidase enzyme function.

### 3.7. Bioactivities of Nanoemulsified 95% Ethanol Extract (95E) of C. nutans 

#### 3.7.1. Particle Size of the Globules in Nanoemulsified 95E Extract of *C. nutans*

The measurement of particle size by NanoSizer is shown in Figure 2. The analysis revealed a mean particle size of 171.93 ± 2.27 nm and a polydispersity index (PDI) of 0.19 ± 0.04, reflecting a uniform particle size distribution. The small particle size and low PDI suggest that the nanoemulsion formulation is stable and well-dispersed, which is critical for enhancing bioavailability and ensuring efficient delivery of the active compounds. Such a narrow size distribution enhances the potential for improved penetration into biofilms and deeper layers of infected tissues, which is essential for maximizing antimicrobial efficacy. The observed particle size falls within the optimal range for nanoemulsion stability, which also enhances its potential for use in therapeutic applications, particularly in overcoming biofilm-associated resistance in oral pathogens.

#### 3.7.2. Anti-Biofilm Formation and Biofilm Degradation

The effectiveness of the nanoemulsified 95E *C. nutans* extract in inhibiting biofilm formation and promoting biofilm degradation in oral pathogens, including *S. mutans*, *S. pyogenes*, *K. pneumoniae*, and *S. aureus,* was evaluated. Both the crude 95E extract and its nanoemulsified form exhibited similar efficacy, achieving over 80% inhibition of biofilm formation (Figure 3A). However, the nanoemulsified formulation at a concentration of 5 mg/mL significantly outperformed the crude extract in inhibiting *S. mutans* biofilm formation. Furthermore, the nanoemulsified extract demonstrated superior biofilm degradation across all bacterial species compared to the crude extract, with particularly notable improvements at the 5 mg/mL concentration (Figure 3B). It is worth mentioning that control lipid nanoparticles devoid of the API showed no significant effect on biofilm formation or degradation; therefore, the results for these control lipid nanoparticles were not included in the graph.

#### 3.7.3. Inhibition of α-Glucosidase Enzyme Activity

This study demonstrated that nanoemulsified extracts significantly enhanced the inhibition of α-glucosidase enzyme activity, with the highest potency achieving approximately eight-fold greater inhibition compared to conventional extracts, with IC50 values of 1.67 ± 0.05 mg/mL and 13.53 ± 0.37 mg/mL, respectively (Figure 4). This highlights the potential of Nanostructured Lipid Carriers (NLCs) as an effective strategy for enhancing the delivery and bioactivity of natural bioactive compounds in diabetes treatment, offering the advantage of reduced drug concentrations and improved therapeutic efficacy. It is important to note that control lipid nanoparticles devoid of the API showed no significant effect on α-glucosidase inhibition; therefore, the results for these control lipid nanoparticles were not included in the graph.

## 4. Discussion

This study investigates the multifaceted bioactivities of *C. nutans* extracts, focusing prominently on the 95% ethanol (95E) extract and its nanoemulsified form, highlighting their potential applications in oral health and beyond. The antibacterial assessments reveal robust efficacy against key oral pathogens: *S. mutans*, *S. pyogenes*, *K. pneumoniae*, and *S. aureus*, showing significant inhibition zones, minimal inhibitory concentrations (MICs) and minimum bactericidal concentrations (MBCs) that underscore their potential in combating periodontal diseases linked to bacterial infections. Moreover, the extracts exhibit notable anti-biofilm properties, crucial for disrupting microbial communities that contribute to plaque formation and dental decay. Previous studies emphasized the anti-biofilm properties of chloroform extracts from *C. nutans* leaves and its isolated compounds, particularly effective against *S. mutans* biofilms [11]. Additionally, research on *Acanthus polystachyus* Delile, a plant from the Acanthaceae family akin to *C. nutans*, revealed that its methanol extract inhibited biofilm formation in *S. aureus* and *S. pyogenes* [21]. This dual action against both planktonic bacteria and biofilms suggests *C. nutans* extracts as promising alternatives or adjuncts to conventional antimicrobial agents in dental care. Furthermore, the extracts demonstrate potent inhibition of α-glucosidase enzyme activity, pivotal in managing diabetes by delaying carbohydrate digestion and subsequent glucose absorption. This study’s findings position *C. nutans* 95E extract as a potent inhibitor, potentially mitigating postprandial hyperglycemia. Previous studies by [22,23] highlighted the variable anti-diabetic effects of *C. nutans* leaf extracts extracted with different solvents, showing that the hexane fraction was particularly effective against α-glucosidase compared to methanol extract. Susanti [24] also reported on the anti-α-glucosidase activity of *C. nutans* extracts, with methanol and ethanol extracts demonstrating significant activity, while the water extract exhibited comparatively lower effectiveness. Beyond their antimicrobial and anti-diabetic roles, the extracts exhibit significant anti-inflammatory effects, reducing nitric oxide production in macrophages stimulated with lipopolysaccharides (LPS). Comparison with literature supports the efficacy of ethanolic extracts in mitigating NO production, aligning with studies by [17,25]. These results underscore the potential of the 95E extract for further exploration as a therapeutic agent with robust anti-inflammatory activity. This anti-inflammatory activity underscores the extracts’ potential in managing inflammatory conditions associated with oral infections and systemic inflammation. The particle size of nanoemulsion is a crucial factor that impacts various key aspects such as stability, encapsulation efficiency, drug release profile, biodistribution, mucoadhesion, and cellular uptake. The size and polydispersity index (PDI) of nanoemulsified *C. nutans* 95E extract were less than 0.7, indicating that the particle size distribution is confined to a narrow range, providing strong evidence of formulation stability [26]. Nanoemulsification further enhances these bioactivities, improving solubility, stability, and bioavailability, thereby enhancing their therapeutic potential. The enhanced anti-biofilm activity observed in nanoparticle-encapsulated extracts may be attributed to favorable nanoparticle properties such as increased loading capacity, improved stability, reduced drug leakage, enhanced bioavailability, and improved permeation enhancement [27]. While silver nanoparticle formulations of *C. nutans* extract have shown antibacterial activity, they may be limited in enhancing the extract’s anti-diabetic and anti-inflammatory properties, potentially due to concerns over prolonged silver ion exposure toxicity. Therefore, employing Nanostructured Lipid Carriers (NLCs) for extract encapsulation could offer broader benefits and mitigate these concerns effectively. NLC are extensively employed in pharmaceutical formulations for oral, topical, and parenteral delivery, primarily due to their superior ability to enhance drug stability and bioavailability as well as being scalable for industrial production compared to other methods. In contrast, electrospraying and electrospinning are less established on an industrial scale, though advances are being made. They are more suited for lab-scale and specialized applications. However, this research has limitations that highlight the need for further studies on the stability of nanoemulsions. In particular, a more detailed investigation into the mechanisms driving inflammation in gum cells is required, along with strategies to improve the specificity of targeting periodontal bacteria. Moreover, it should be noted that while the nanoemulsified *C. nutans* extracts demonstrated enhanced efficacy within a short period (15 min for α-glucosidase inhibition and 24 h for biofilm formation inhibition), the stability of these nanoemulsions requires further investigation. As a limitation of the current study, the long-term stability and sustained efficacy of the nanoemulsions have not been explored. Future research should focus on assessing the stability of the nanoemulsions over extended periods to better understand their therapeutic potential in clinical settings. The synergistic benefits of nanoemulsified *C. nutans* extracts suggest promising avenues for future research and development of novel therapeutic agents targeting oral health and metabolic disorders.

## 5. Conclusions

In this study, *Clinacanthus nutans* extracts (95M, 95E, 70E, and W) demonstrated antibacterial, anti-biofilm, anti-inflammatory, and anti-α-glucosidase activities, with the 95% ethanolic (95E) extract and its nanoemulsified form that showed enhanced anti-biofilm and anti-α-glucosidase effects. The extract exhibited strong antibacterial efficacy against key oral pathogens, *S. mutans* and *S. aureus*. Notably, the significant anti-biofilm activity of the extract is crucial for mitigating dental plaque formation. Additionally, these extracts effectively inhibited the α-glucosidase enzyme, suggesting potential therapeutic benefits for diabetes management by regulating glucose levels and controlling the growth of oral bacteria. Their anti-inflammatory properties, evidenced by reduced nitric oxide production, further highlight their potential in treating oral infections and inflammatory conditions. Interestingly, the enhanced effects observed with the nanoemulsified 95E extract emphasize its potential for managing periodontal disease in diabetic patients. Future research should focus on elucidating the mechanisms driving the anti-inflammatory activity in gum cells and evaluating the stability of these extracts prior to validating their clinical efficacy and safety profiles.

## Figures and Tables

**Figure 1 biology-13-00815-f001:**
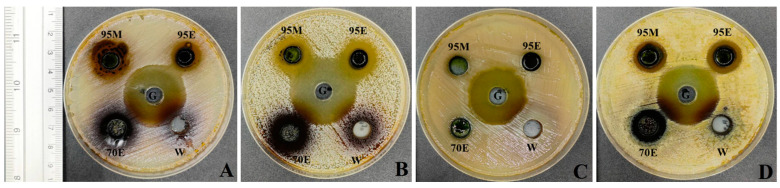
Antibacterial activity of *C. nutans* extracts assessed by the agar-well diffusion method. (**A**) *S. mutans*, (**B**) *S. pyogenes*, (**C**) *K. pneumoniae*, and (**D**) *S. aureus*, with G = gentamicin as the positive control. Each test was performed using different extracts: 95% methanol (95M), 95% ethanol (95E), 70% ethanol (70E), and water (W).

**Figure 2 biology-13-00815-f002:**
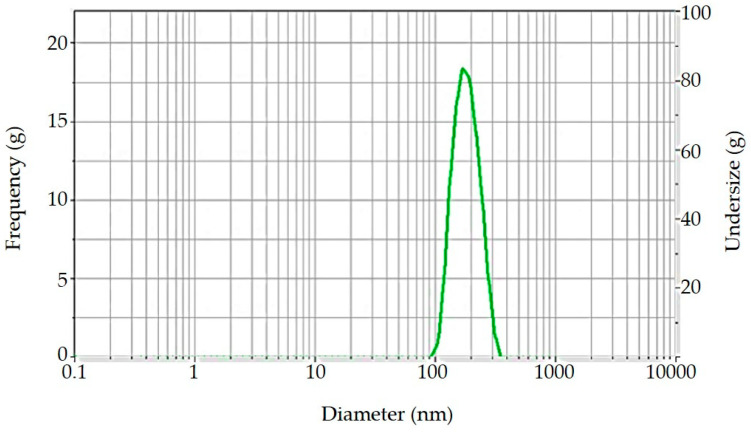
Particle size distribution of nanoemulsified 95E extract of *C. nutans* measured using NanoSizer.

**Figure 3 biology-13-00815-f003:**
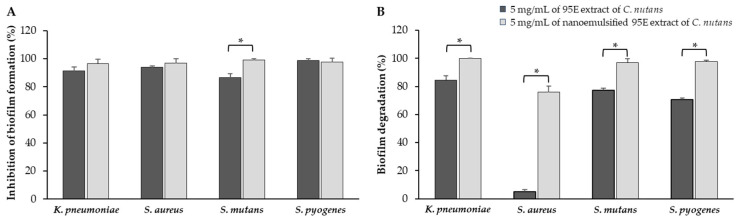
Comparative analysis of biofilm inhibition: (**A**) biofilm formation and (**B**) biofilm degradation by 95E extract of *C. nutans* and its nanoemulsified form. Data are presented as mean ± SD from three independent experiments. Statistical significance is denoted by * *p* < 0.001, indicating significant differences between extract and nanoemulsion formulations.

**Figure 4 biology-13-00815-f004:**
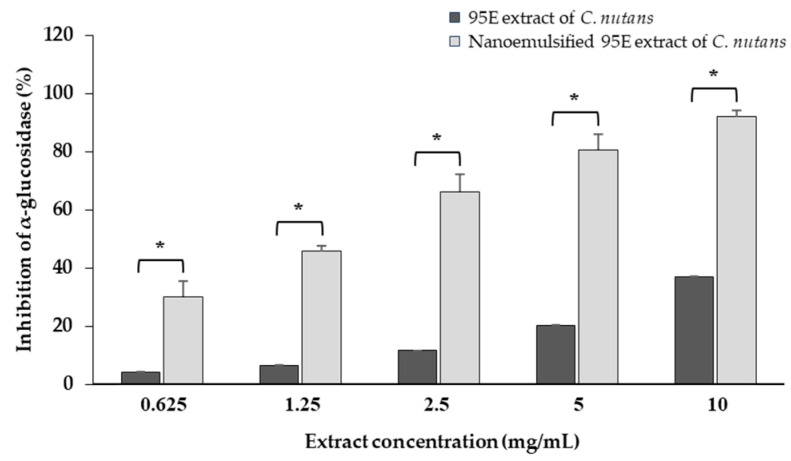
Comparison of α-glucosidase inhibition activity between 95E *C. nutans* extract and its nanoemulsified formulation. * *p* < 0.001 indicates a significant difference between the 95E extract and the nanoemulsified 95E.

**Table 1 biology-13-00815-t001:** Antibacterial efficacy of *C. nutans* extracts against *S. mutans*, *S. pyogenes*, *K. pneumoniae*, and *S. aureus*, evaluated by the agar-well diffusion (500 mg/mL) and MIC/MBC assays.

Bacterial Species	*C. nutans* Extracts	Inhibition Zone Diameter (mm)	MIC and MBC (mg/mL)
*S. mutans*	95M	16.67 ± 0.58 ^b^	125 ± 0.00 ^c^
	95E	17.00 ± 0.00 ^b^	250 ± 0.00 ^b^
	70E	16.00 ± 1.00 ^b^	125 ± 0.00 ^c^
	W	10.00 ± 0.00 ^c^	>250 ± 0.00 ^a^
	Gentamicin	32.33 ± 0.58 ^a^	0.008 ± 0.00 ^d^
*S. pyogenes*	95M	16.67 ± 2.31 ^b^	250 ± 0.00 ^b^
	95E	19.67 ± 0.58 ^b^	250 ± 0.00 ^b^
	70E	17 ± 1.00 ^b^	250 ± 0.00 ^b^
	W	10 ± 0.00 ^c^	>250 ± 0.00 ^a^
	Gentamicin	34 ± 1.00 ^a^	>0.5 ± 0.00 ^c^
*K. pneumoniae*	95M	11.33 ± 0.58 ^b^	250 ± 0.00 ^b^
	95E	11.67 ± 0.58 ^b^	250 ± 0.00 ^b^
	70E	11.00 ± 0.00 ^b^	250 ± 0.00 ^b^
	W	10.00 ± 0.00 ^c^	>250 ± 0.00 ^a^
	Gentamicin	30.00 ± 0.00 ^a^	0.0005 ± 0.00 ^c^
*S. aureus*	95M	15.33 ± 1.15 ^b^	125 ± 0.00 ^c^
	95E	16.33 ± 1.15 ^b^	125 ± 0.00 ^c^
	70E	17.00 ± 0.00 ^b^	250 ±0.00 ^b^
	W	10.00 ± 0.00 ^c^	>250 ± 0.00 ^a^
	Gentamicin	29.67 ± 0.58 ^a^	0.004 ± 0.00 ^d^

Note: The wells were 10 mm in diameter. The data are expressed as mean ± S.D. (*n* = 3). Statistically significant differences were found among the *C. nutans* extracts (*p* < 0.05), with the values ranked in order of significance as follows: a, b, c, and d.

**Table 2 biology-13-00815-t002:** Anti-biofilm activity of *C. nutans* extracts on biofilm formation and degradation.

Bacterial Species	*C. nutans* Extracts	Concentration (mg/mL)	Inhibition of Biofilm Formation (%)	Degradation of Biofilm (%)
*S. mutans*	95M	62.5	81.05 ± 0.66 ^b^	82.18 ± 2.72 ^b^
	95E	125	97.36 ± 3.93 ^a^	99.32 ± 1.08 ^a^
	70E	62.5	59.25 ± 1.24 ^c^	12.76 ± 2.59 ^d^
	W	250	82.03 ± 1.35 ^b^	80.23 ± 1.75 ^b^
	Gentamicin	0.004	32.46 ± 3.09 ^d^	23.50 ± 2.45 ^c^
*S. pyogenes*	95M	125	75.03 ± 2.14 ^b^	99.40 ± 0.98 ^a^
	95E	125	99.86 ± 0.25 ^a^	99.23 ± 0.74 ^a^
	70E	125	16.49 ± 1.11 ^e^	97.07 ± 1.19 ^a^
	W	250	54.84 ± 3.36 ^c^	55.98 ± 0.31 ^b^
	Gentamicin	0.5	26.11 ± 0.45 ^d^	37.47 ± 2.16 ^c^
*K. pneumoniae*	95M	125	98.13 ± 1.41 ^a^	71.56 ± 2.40 ^c^
	95E	125	84.78 ± 0.98 ^bc^	84.15 ± 2.59 ^b^
	70E	125	71.15 ± 1.40 ^d^	97.32 ± 1.18 ^a^
	W	250	85.74 ± 1.81 ^b^	83.49 ± 0.39 ^b^
	Gentamicin	0.00025	80.68 ± 2.32 ^c^	87.73 ± 1.04 ^b^
*S. aureus*	95M	62.5	93.12 ± 1.51 ^bc^	77.08 ± 1.82 ^ab^
	95E	62.5	99.79 ± 0.37 ^a^	84.13 ± 4.54 ^a^
	70E	125	100.00 ± 0.00 ^a^	35.40 ± 1.84 ^d^
	W	250	95.44 ± 2.30 ^b^	72.12 ± 6.12 ^bc^
	Gentamicin	0.004	90.60 ± 0.65 ^c^	64.90 ± 1.34 ^c^

Note: The data are expressed as mean ± S.D. (*n* = 3). Statistically significant differences were found among the *C. nutans* extracts (*p* < 0.05), with the values ranked in order of significance as follows: a, b, c, d, and e.

**Table 3 biology-13-00815-t003:** Effect of *C. nutans* extracts on inhibition of α-glucosidase enzyme.

*C. nutans* Extracts	IC50 (mg/mL)
95M	12.97 ± 0.55 ^d^
95E	17.19 ± 0.30 ^c^
70E	7.57 ± 0.04 ^e^
W	23.16 ± 1.33 ^b^
Acarbose	48.64 ± 0.43 ^a^

Note: The data are expressed as mean ± S.D. (*n* = 3). Statistically significant differences were found among the *C. nutans* extracts (*p* < 0.05), with the values ranked in order of significance as follows: a, b, c, d, and e.

**Table 4 biology-13-00815-t004:** Cell viability (%) of *C. nutans* extracts on RAW264.7 macrophage cells.

Extract Concentration (mg/mL)	Cell Viability (%)			
	95M	95E	70E	W
Cell control (CC)	100.00 ± 0.00	100.00 ± 0.00	100.00 ± 0.00	100.00 ± 0.00
Vehicle control	88.86 ± 4.87	88.86 ± 4.87	88.86 ± 4.87	98.99 ± 0.24
0.08	105.09 ± 4.24	103.25 ± 2.60	102.23 ± 1.30	98.69 ± 0.81
0.16	104.75 ± 3.32	104.29 ± 5.14	101.56 ± 1.88	101.27 ± 1.15
0.31	107.83 ± 2.27	105.86 ± 5.49	103.72 ± 3.33	98.26 ± 1.84
0.63	109.25 ± 3.00	105.39 ± 2.86	100.09 ± 2.86	93.98 ± 5.13
1.25	106.74 ± 3.67	97.88 ± 4.54	97.23 ± 0.75	78.29 ± 6.60
2.50	101.01 ± 5.87	73.54 ± 4.48	81.38 ± 1.13	72.19 ± 6.98
5.00	54.81 ± 6.84	1.87 ± 0.51	47.22 ± 9.43	69.61 ± 4.74

Note: Data are mean ± S.D. (*n* = 3). Moreover, 1% of dimethyl sulfoxide (DMSO) and water were vehicle controls.

**Table 5 biology-13-00815-t005:** IC50 values of *C. nutans* extracts on cell viability of RAW264.7 macrophage cells.

* C. nutans * Extracts	IC50 (mg/mL)
95M	5.51 ± 0.56
95E	3.07 ± 0.05
70E	5.15 ± 0.26
W	3.75 ± 1.12

Note: Data are mean ± S.D. (*n* = 3).

**Table 6 biology-13-00815-t006:** Inhibition of nitric oxide (NO) secretion by *C. nutans* extracts in stimulated RAW264.7 macrophage culture cells.

Extracts Concentration (mg/mL)	Inhibition of NO Secretion (%)			
	95M	95E	70E	W
0.08	31.90 ± 1.80	41.47 ± 2.05	31.35 ± 2.09	0.00 ± 0.00
0.16	74.84 ± 2.02	67.38 ± 3.79	58.14 ± 5.55	0.00 ± 0.00
0.31	94.88 ± 1.33	88.22 ± 2.81	82.82 ± 1.99	0.00 ± 0.00
0.63	104.32 ± 1.64	92.61 ± 3.00	103.77 ± 5.10	0.00 ± 0.00
1.25	114.04 ± 2.64	94.17 ± 4.33	113.91 ± 5.39	0.00 ± 0.00
2.5	125.43 ± 4.58	97.65 ± 4.41	125.19 ± 3.15	6.61 ± 4.98

Note: Data are mean ± S.D. (*n* = 3).

**Table 7 biology-13-00815-t007:** IC50 value of *C. nutans* extracts in inhibiting NO secretion from stimulated RAW264.7 macrophage culture cells.

* C. nutans * Extracts	IC50 (mg/mL)
95M	0.10 ± 0.03
95E	0.10 ± 0.00
70E	0.15 ± 0.02
W	>2.5

Note: Data are mean ± S.D. (*n* = 3).

## Data Availability

All data supporting the findings of this study are included in the main manuscript.

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
