# Peer review of "Multifunctional Nanoemulsified Clinacanthus nutans Extract: Synergistic Anti-Pathogenic, Anti-Biofilm, Anti-Inflammatory, and Metabolic Modulation Effects against Periodontitis"

_biology, 2024, doi:10.3390/biology13100815_

Round 1
Reviewer 1 Report
Comments and Suggestions for Authors
The present study investigates the Synergistic Anti-Pathogenic, Anti-Biofilm, Anti-Inflammatory, and Metabolic Modulation Effects against Periodontitis of a multifunctional Nanoemulsified Clinacanthus nutans Extract. After a careful examination, the reviewer found the work not ready for publication. Please consider the following observations:
Lines 133-136: The extracts, labeled as 95M, 95E, 70E, and W according to their respective extraction solvents, were stored in a frozen state until further use. For cell treatment applications, the extracts were dissolved in DMSO. The authors omit the evaluation of the bioactivity of DMSO.
The authors do not include evidence of nanoemulsion formation (characterization of nanoemulsion formation MUST BE INCLUDED).
The authors do not include graphical evidence (images of bacterial growth inhibition) of their results.
Comments on the Quality of English LanguageModerate editing of English grammar
Author Response
Comment 1: Lines 133-136: The extracts, labeled as 95M, 95E, 70E, and W according to their respective extraction solvents, were stored in a frozen state until further use. For cell treatment applications, the extracts were dissolved in DMSO. The authors omit the evaluation of the bioactivity of DMSO.
Response 1: Thank you for pointing this out, in this experiment, we excluded the evaluation of bioactivity, such as the antibacterial activity of DMSO because numerous reports have indicated that DMSO lacked such properties such as the publication of Kebede et al. (2021). Therefore, any observed antibacterial effects can be attributed solely to the active ingredients in the extract, rather than to DMSO. However, in the study of cell viability with RAW264.7 cells, the toxicity of DMSO was included to confirm that it did not induce cell death as shown in Table 4.
Comment 2 The authors do not include evidence of nanoemulsion formation (characterization of nanoemulsion formation MUST BE INCLUDED).
Response 2: Thank you for your comment. We have provided additional details regarding the particle size and polydispersity index (PDI) of the nanoemulsion. The information is included in Line 479 as follows
“The size and polydispersity index (PDI) of nanoemulsified C. nutans 95E extract were less than 0.7, indicating that the particle size distribution is confined to a narrow range, providing strong evidence of formulation stability [26].”
Comment 3 The authors do not include graphical evidence (images of bacterial growth inhibition) of their results.
Response 3: Thank you for your comment. We have included images showing bacterial growth inhibition (clear zones of inhibition) obtained through the agar well diffusion method in Figure 1 (Line 300) as follows.
Figure 1 Antibacterial activity of C. nutans extracts assessed by agar well diffusion method. (A) S. mutans, (B) S. pyogenes, (C) K. pneumoniae, (D) S. aureus, with G = gentamicin as the positive control. Each test was performed using different extracts: 95% methanol (95M), 95% ethanol (95E), 70% ethanol (70E), and water (W).

Reviewer 2 Report
Comments and Suggestions for Authors
The manuscript reports that Clinacanthus nutans extracts from the 95% ethanol had fine antibacterial efficacy against key oral pathogens for potential treatments of periodontal diseases. The topic is interesting. The contents are new and fall well within the scope of this important journal. I recommend its acceptance for publication after minor revision.
The DISCUSSION section is too small. It can be expanded from several directions, e.g. the present preparation of nanostructured lipid carriers (NLC) utilizing hot and high-pressure homogenization techniques, can be compared with the electrospraying ( https://doi.org/10.3390/gels9090700) and electrospinning (Macromol. Mater. Eng. 2023, 309, 2300361 & https://doi.org/10.1016/j.pmatsci.2024.101350), which have been broadly exploited to load plant extracts and to load and manipulate controlled release profiles of numerous poorly water-soluble drugs.
Conclusion section can be rephrased according to the following items: A summary of your findings; A synopsis of your new concepts and innovations; A brief restatement of your hypotheses; A comparison with findings by other workers and your vision for future work.
Please pay attention to the references’ formats according to the requests of journal, particularly, there are several unexpected blanks.
Author Response
Comment 1: The DISCUSSION section is too small. It can be expanded from several directions, e.g. the present preparation of nanostructured lipid carriers (NLC) utilizing hot and high-pressure homogenization techniques, can be compared with the electrospraying (https://doi.org/10.3390/gels9090700) and electrospinning (Macromol. Mater. Eng. 2023, 309, 2300361 & https://doi.org/10.1016/j.pmatsci.2024.101350), which have been broadly exploited to load plant extracts and to load and manipulate controlled release profiles of numerous poorly water-soluble drugs.
Response 1: Thank you for your comment, we agree that the discussion could be further elaborated. We revised the discussion accordingly addressing all points and suggestions raised by the reviewer as shown below.
Line 483
“The enhanced anti-biofilm activity observed in nanoparticle-encapsulated extracts may be attributed to favorable nanoparticle properties such as increased loading capacity, improved stability, reduced drug leakage, enhanced bioavailability, and improved permeation enhancement [27]. While silver nanoparticle formulations of C. nutans extract have shown antibacterial activity, they may be limited in enhancing the extract's anti-diabetic and anti-inflammatory properties, potentially due to concerns over prolonged silver ion exposure toxicity. Therefore, employing Nanostructured Lipid Carriers (NLCs) for extract encapsulation could offer broader benefits and mitigate these concerns effectively. NLC are extensively employed in pharmaceutical formulations for oral, topical, and parenteral delivery, primarily due to their superior ability to enhance drug stability and bioavailability as well as scalable for industrial production compared to other methods. In contrast, electrospraying and electrospinning are less established on an industrial scale, though advances are being made. They are more suited for lab-scale and specialized applications. However, this research has limitations that highlight the need for further studies on the stability of nanoemulsions. In particular, a more detailed investigation into the mechanisms driving inflammation in gum cells is required, along with strategies to improve the specificity of targeting periodontal bacteria.”
Comment 2: Conclusion section can be rephrased according to the following items: A summary of your findings; A synopsis of your new concepts and innovations; A brief restatement of your hypotheses; A comparison with findings by other workers and your vision for future work.
Response 2: Thank you for your comment. We have made changes accordingly to restructure the conclusion section (Line 512) as follows.
“In this study, Clinacanthus nutans extracts (95M, 95E, 70E, and W) demonstrated antibacterial, anti-biofilm, anti-inflammatory, and anti-diabetic activities, with the 95% ethanolic (95E) extract and its nanoemulsified form that showed enhanced anti-biofilm and anti-α-glucosidase effects. The extract exhibited strong antibacterial efficacy against key oral pathogens, S. mutans and S. aureus. Notably, the significant anti-biofilm activity of the extract is crucial for mitigating dental plaque formation. Additionally, these extracts effectively inhibited the α-glucosidase enzyme, suggesting potential therapeutic benefits for diabetes management by regulating glucose levels and controlling the growth of oral bacteria. Their anti-inflammatory properties, evidenced by reduced nitric oxide production, further highlight their potential in treating oral infections and inflammatory conditions. Interestingly, the enhanced effects observed with the nanoemulsified 95E extract emphasize its potential for managing periodontal disease in diabetic patients. Future research should focus on elucidating the mechanisms driving the anti-inflammatory activity in gum cells and evaluating the stability of these extracts, prior to validating their clinical efficacy and safety profiles.”
Comment 3: Please pay attention to the references’ formats according to the requests of journal, particularly, there are several unexpected blanks.
Response 3: Thank you for your comment and apologies for the mistakes. We have rechecked and revised all references as shown in the reference section now Line 548.

Reviewer 3 Report
Comments and Suggestions for Authors
Clinacanthus nutans is a plant endemic to Southeast Asia. In traditional medicine, it has been used to treat numerous ailments, including snakebites and cancer. Due to its undiscovered full therapeutic potential, it has attracted the attention of researchers. The authors of this study investigated the possibility of using various extracts obtained from the leaves of this plant to treat oral infections and inflammation. The study showed that the nanoemulsified 95E extract showed eight times the efficacy of the regular extract.
My comments:
1. Editorial work written in an unattentive manner - not in accordance with journal requirements, e.g.:
- Keywords should be separated by semicolons and not commas
- References should be numbered in order of appearance and indicated by a numeral or numerals in square brackets - e.g., [1] or [2,3], or [4–6]. Instead, the authors give the name of the first author of the publication and the year of publication.
- The text uses different line spacing (lines 286-515).
- Section: Materials and Methods
Subheadings should be in italics and numbered accordingly.
For some of the equipment and materials used, the manufacturer's name and country of origin are missing in brackets.
- Section: Results
Subheadings should be in italics and numbered accordingly.
- Literature references should be written in accordance with the journal's requirements: (Author 1, A.B.; Author 2, C.D. Title of the article. Abbreviated Journal Name Year, Volume, page range.).
- Not very readable: Graphical Abstract, Figure 1and 2.
2. No literature items cited in the text in the References section.
(for example: Steigmann et al., 2022; Wanikiat et al., 2008; Timpawat and Vajrabhaya, 2013; Gross and Wolin, 1995; Matsuno et al., 1998; Shweash et al., 2011
3. No citation in the text:
- Lim SHE, Almakhmari MA, Alameri SI, Chin SY, Abushelaibi A, Mai CW, Lai KS. 2020. Antibacterial activity of 563 Clinacanthus nutans polar and non-polar leaves and stem extracts. Biomedical and Pharmacology Journal 13: 1169-1174.
- Mat Yusuf SNA, Che Mood CNA, Ahmad NH, Sandai D, Lee CK, Lim V. 2020. Optimization of biogenic synthesis of 568 silver nanoparticles from flavonoid-rich Clinacanthus nutans leaf and stem aqueous extracts. Royal Society Open 569 Science 7(7): 200065.
- Sulaiman ISC, Basri M, Masoumi HRF, Ashari SE, Ismail M. 2016. Design and development of a nanoemulsion system 589 containing extract of Clinacanthus nutans (L.) leaves for transdermal delivery system by D-optimal mixture design and 590 evaluation of its physicochemical properties. RSC Advances 6(71): 67378-67388.
- Waiezi S, Malek NANN, Asraf MH, Sani NS. 2023. Preparation, characterization, and antibacterial activity of green-598 biosynthesised silver nanoparticles using Clinacanthus nutans extract. Biointerface Research in Applied Chemistry 13: 171.
4. The introduction does not provide sufficient information on the subject of the manuscript.
Which main phytochemical constituents are present in the dried leaf powder of C. nutans?
- Which active substances were isolated depending on the extraction conditions and solvent used?
5. Line 132: The lyophilisation conditions of the crude extracts would need to be stated.
6. Line 132: How was the percentage yield calculated?
7. What do the indexes: a, b, c, d in Tables 1-3 mean? According to the authors, compared to what was a statistically significant difference observed?
8. Table 4: Please check the accuracy of the results presented:
Was the cell viability (%) of RAW264.7 for extract 95E at a concentration of 5.00 mg/ml 1.87±0.51?
9. Section: Preparation of nanoemulsion of the 95% ethanol extract of C. nutans (95E)
There is insufficient information on the quantitative and qualitative composition of the nanoemulsion and how it was prepared, e.g:
Line 251: Which surfactant and dissolving agents were used?
Line 252: How was the mixture mixed?
Line 253: How much surfactant and water was used to make the aqueous phase?
Kindly correct.
10. In the Results section, the size distribution of the globules in the nanoemulsion should be shown.
11. Has the stability of the nanoemulsion been tested?
12. In the Discussion section, there is no reference of the results obtained by the authors to other studies.
What were the limitations of the study?
Kindly correct.
Author Response
Comment 1: 1. Editorial work written in an unattentive manner - not in accordance with journal requirements, e.g.:
Response 1: Thank you for your evaluation of our manuscripts and apologies that some mistakes remained in the manuscripts. We have revised all points carefully and response to each point as shown below.
- Keywords should be separated by semicolons and not commas
Response: Many thank for your comment, we have revised the manuscript accordingly as shown in Line 44 as follows.
“anti-oral plaque; diabetic periodontal disease; gum healing; multifunctional extract; nanoemulsion delivery; oral pathogenic bacteria”
- References should be numbered in order of appearance and indicated by a numeral or numerals in square brackets - e.g., [1] or [2,3], or [4–6]. Instead, the authors give the name of the first author of the publication and the year of publication.
Response: Many thanks for your comment and apologies for the mistakes. We have revised and rechecked all references accordingly as now shown in Line 548.
- The text uses different line spacing (lines 286-515).
Response: Many thanks for your comment and apologies for the mistakes. We have checked the whole manuscript thoroughly to make sure the line space is equal.
- Section: Materials and Methods
Subheadings should be in italics and numbered accordingly.
Response: Many thanks for your comment we have italicized and numbered all subheadings in materials and methods section accordingly.
For some of the equipment and materials used, the manufacturer's name and country of origin are missing in brackets.
Response: Thank you for pointing this out, we have gone through the materials and methods section and included all information where appropriate as shown below.
Line 125
“The finely ground dried leaf powder of C. nutans was obtained from Lampang Herbs Shop in Lampang Province, (Lampang Herbs Shop, Thailand).”
Line 132
“rotary evaporator (Rotavapor® R-300, BÜCHI Labortechnik AG, Switzerland)”
Line 134
“freeze dryer (Lyovapor™ L-200 Freeze Dryer. BÜCHI Labortechnik AG, Switzerland)”
- Section: Results
Subheadings should be in italics and numbered accordingly.
Response: Many thanks for your comment we have italicized and numbered all subheadings in results section accordingly.
- Literature references should be written in accordance with the journal's requirements: (Author 1, A.B.; Author 2, C.D. Title of the article. Abbreviated Journal Name Year, Volume, page range.).
Response: Thank you for your comment, we have revised the manuscript accordingly.
- Not very readable: Graphical Abstract, Figure 1 and 2.
Response: Thank you for your comment, We have improved the quality of the Graphical Abstract, Figure 1 (now revised as Figure 3), and Figure 2 (now revised as Figure 4) as suggested.
Graphical Abstract
Figure 3 Comparative analysis of biofilm inhibition: (A) biofilm formation and (B) biofilm degradation by 95E extract of C. nutans and its nanoemulsified form. Data are presented as mean ± SD from three independent experiments. Statistical significance is denoted by *p<0.001, indicating significant differences between extract and nanoemulsion formulations.
Figure 4 Comparison of α-glucosidase inhibition activity between 95E C. nutans extract and its nanoemulsified formulation. *p < 0.001 indicates a significant difference between the 95E extract and the nanoemulsified 95E.
Comment 2: No literature items cited in the text in the References section. (for example: Steigmann et al., 2022; Wanikiat et al., 2008; Timpawat and Vajrabhaya, 2013; Gross and Wolin, 1995; Matsuno et al., 1998; Shweash et al., 2011
Response 2: Thank you for pointing this out, we have rechecked all references and revised the missing references in the main text accordingly. The cited references are as listed below.
- Steigmann, L.; Maekawa, S.; Kauffmann, F.; Reiss, J.; Cornett, A.; Sugai, J.; Lombaert, I.M. Changes in salivary biomarkers associated with periodontitis and diabetic neuropathy in individuals with type 1 diabetes. Scientific Reports 2022, 12(1), 11284.
- Wanikiat, P.; Panthong, A.; Sujayanon, P.; Yoosook, C.; Rossi, A.G.; Reutrakul, V. The anti-inflammatory effects and the inhibition of neutrophil responsiveness by Barleria lupulina and Clinacanthus nutans Journal of Ethnopharmacology 2008, 116(2), 234-244.
- Timpawat, S., Vajrabhaya, L.O. The efficacy of Clinacanthus nutans in the treatment of recurrent aphthous stomatitis: A double-blind controlled trial. Journal of Oral Pathology & Medicine 2013, 42(1), 56–60.
- Gross, S.S.; Wolin, M.S. Nitric oxide: pathophysiological mechanisms. Annual Review of Physiology 1995, 57(1), 737–769.
- Matsuno, K.; Eastman, D.; Mitsiades, T.; Quinn, A.M.; Carcanciu, M.L.; Ordentlich, P.; Artavanis-Tsakonas, S. Human deltex is a conserved regulator of Notch signaling. Nature Genetics 1998, 19(1), 74-78.
- Shweash, M.; McGachy, H.A.; Schroeder, J.; Neamatallah, T.; Bryant, C.E.; Millington, O.; Plevin, R. Leishmania mexicana promastigotes inhibit macrophage IL-12 production via TLR-4 dependent COX-2, iNOS and arginase-1 expression. Molecular Immunology 2011, 48(15-16), 1800–1808.
Comment 3: 3. No citation in the text:
- Lim SHE, Almakhmari MA, Alameri SI, Chin SY, Abushelaibi A, Mai CW, Lai KS. 2020. Antibacterial activity of 563 Clinacanthus nutans polar and non-polar leaves and stem extracts. Biomedical and Pharmacology Journal 13: 1169-1174.
- Mat Yusuf SNA, Che Mood CNA, Ahmad NH, Sandai D, Lee CK, Lim V. 2020. Optimization of biogenic synthesis of 568 silver nanoparticles from flavonoid-rich Clinacanthus nutans leaf and stem aqueous extracts. Royal Society Open 569 Science 7(7): 200065.
- Sulaiman ISC, Basri M, Masoumi HRF, Ashari SE, Ismail M. 2016. Design and development of a nanoemulsion system 589 containing extract of Clinacanthus nutans (L.) leaves for transdermal delivery system by D-optimal mixture design and 590 evaluation of its physicochemical properties. RSC Advances 6(71): 67378-67388.
- Waiezi S, Malek NANN, Asraf MH, Sani NS. 2023. Preparation, characterization, and antibacterial activity of green-598 biosynthesised silver nanoparticles using Clinacanthus nutans extract. Biointerface Research in Applied Chemistry 13: 171.
Response 3: Thank you for bringing this to our attention. We have carefully reviewed the manuscript and confirmed that the mentioned references were intentionally omitted from the revised version. These references were not relevant to the final scope and focus of the manuscript, and thus were not included. We appreciate your understanding and have ensured that all included references accurately reflect the content of the revised manuscript.
Comment 4: The introduction does not provide sufficient information on the subject of the manuscript.
Response 4: Thank you for your comment, we have provided additional information about the C. nutans plant and the bioactive ingredients in the dried leaf powder of C. nutans in the introduction Line 72 as follows.
“Clinacanthus nutans (Burm.f.) Lindau, a medicinal plant rich in bioactive compounds exhibit diverse pharmacological properties, including analgesic, anti-inflammatory, antiviral, antibacterial, anti-biofilm, antioxidant, protect against free radical-induced hemolysis, immunomodulatory, and antidiabetic properties [7-13]. This plant is listed in Thai Herbal Pharmacopoeias with reported uses of crushed leaves soaked in 40% ethanol as anti-viral and anti-inflammatory agents. The dried leaf powder of C. nutans contains various phytochemical constituents, including phenolic compounds, flavonoids, alkaloids, saponins, triterpenoids, diterpenes, steroids, phytosterols, tannins, carbohydrates, and proteins/amino acids [14].”
Comment 5: Line 132: The lyophilisation conditions of the crude extracts would need to be stated.
Response 5: Many thanks for your comment, we have provided additional information in the method section Line 133 as follows.
“The crude extracts were lyophilized by freeze dryer (Lyovapor™ L-200 Freeze Dryer. BÜCHI Labortechnik AG, Switzerland) until dryness. The lyophilization process in-volves freezing the crude extract at -80 °C for 12 hours, followed by a primary drying phase, where the temperature is gradually raised to -20 °C under vacuum for 24 hours, allowing sublimation to occur. Finally, the secondary drying phase involves increasing the temperature to 10 °C for an additional 12 hours to remove any remaining bound water. The entire freeze-drying process takes 48 hours. The weight of the lyophilized extracts were measured and calculate the percentage yield of each extract shown as the equation below.”
Comment 6: Line 132: How was the percentage yield calculated?
Response 6: Thank you for your comment, we agree that the equation used to calculate the percentage yield of each extract should be included. The equation is now presented in Line 146 as follows.
“Percentage Yield = (Final weight of extract/ Initial weight of extract) × 100”
Comment 7: What do the indexes: a, b, c, d in Tables 1-3 mean? According to the authors, compared to what was a statistically significant difference observed?
Response 7: Thank you for pointing this out, we have provided explanations for the details shown in the footnotes of the tables (Line 335, 369, 373). The example of the revised footnotes is shown below
Table 1 “Note. The data are expressed as mean ± SD (n = 3). Statistically significant differences were found among the C. nutans extracts (p < 0.05), with the values ranked in order of significance as follows: a, b, c, and d.”
Table 2 “Note: The data are expressed as mean ± SD (n = 3). Statistically significant differences were found among the C. nutans extracts (p < 0.05), with the values ranked in order of significance as follows: a, b, c, d, and e.”
Table 3 “Note: The data are expressed as mean ± SD (n = 3). Statistically significant differences were found among the C. nutans extracts (p < 0.05), with the values ranked in order of significance as follows: a, b, c, d, and e.”
Comment 8: Table 4: Please check the accuracy of the results presented: Was the cell viability (%) of RAW264.7 for extract 95E at a concentration of 5.00 mg/ml 1.87±0.51?
Response 8: Thank you for your comment, we have double-checked the experimental results and we ensured that the values presented were correct and this was the result from three independent experiments reported with standard deviation (SD).
Comment 9: Section: Preparation of nanoemulsion of the 95% ethanol extract of C. nutans (95E)
There is insufficient information on the quantitative and qualitative composition of the nanoemulsion and how it was prepared, e.g:
Line 251: Which surfactant and dissolving agents were used?
Line 252: How was the mixture mixed?
Line 253: How much surfactant and water was used to make the aqueous phase?
Kindly correct.
Response 9: Thank you for your comment, we agree that more details on the preparation should be included. We have revised this section to add more information about the types of components used in the preparation of the nanoemulsion, along with the proportions of each substance expressed as percentages in Line 259 as shown below.
“2.7 Preparation of nanoemulsion of the 95% ethanol extract of C. nutans (95E)
The preparation of nanostructured lipid carriers (NLC) utilizing hot and high-pressure homogenization techniques began with the preparation of the oil phase by weighing the MCT oil (20%), 95E extract (1%), span 80 (12%), and B-wax (2%). The mixture was stirred at a speed of 300 rpm while being heated to 75°C, ensuring all components dissolved thoroughly to form a uniform mixture. Next, the aqueous phase was prepared by weighing the tween 20 (3%), glycerin (2%), poloxamer 188 (2%), phenoxyethanol (0.1%) and adding water to 100%. This solution was agitated at a speed of 300 rpm while being heated to 75°C, ensuring complete dissolution of all components into a uniform mixture. Subsequently, the aqueous phase was introduced into the oil phase while mechanically stirring at 300 rpm for 20 minutes at 75°C to form a pre-emulsion. The nanoparticles were further reduced in size by employing a homogenizer for a duration of 10 minutes. This process was carried out to ensure the resulting solution's homogeneity and achieve the desired nanoparticle size. Therefore, the size of nanoemulsion was investigated by NanoSizer (nanoparticle size must not exceed 200 nm). Subsequently, the biological activities of the nanoemulsion were evaluated.”
Comment 10: In the Results section, the size distribution of the globules in the nanoemulsion should be shown.
Response 10: Many thanks for your comment, we agree that the size distribution of the globules should be presented to comprehensively provide the information to the readers. We have added a graph depicting the size distribution of the nanoemulsion, as measured by the NanoSizer as Figure 2 shown below.
Figure 2 Particle size distribution of nanoemulsified 95E extract of C. nutans measured using NanoSizer.
Comment 11: Has the stability of the nanoemulsion been tested?
Response 11: Thank you for your question, we agree that the stability of the nanoemulsion is another aspect that could be further investigated, however, in this work, we have yet to test for the stability of our nanoemulsion. To ensure that this could be the very next step, we have suggested this in the discussion part along with the reports of previous works (Line 500) as shown below.
“Moreover, it should be noted that while the nanoemulsified C. nutans extracts demonstrated enhanced efficacy within a short period (15 minutes for α-glucosidase inhibition and 24 hours for biofilm formation inhibition), the stability of these nanoemulsions requires further investigation. As a limitation of the current study, the long-term stability and sustained efficacy of the nanoemulsions have not been explored. Future research should focus on assessing the stability of the nanoemulsions over extended periods to better understand their therapeutic potential in clinical settings.”
Comment 12: In the Discussion section, there is no reference of the results obtained by the authors to other studies.
Response 12: The authors have relocated the meta-discussions that were interspersed with some of the experimental results to the dedicated experimental meta-discussion section. They have also expanded this section by incorporating references to other studies in Line 454 as follows.
“Previous studies emphasized the anti-biofilm properties of chloroform extracts from C. nutans leaves and its isolated compounds, particularly effective against S. mutans biofilms [11]. Additionally, research on Acanthus polystachyus Delile, a plant from the Acanthaceae family akin to C. nutans, revealed that its methanol extract inhibited biofilm formation in S. aureus and S. pyogenes [21].”
Comment 13: What were the limitations of the study?
Response 13: Thank you for your question, we agree that the limitations of this study should be mentioned to comprehend the study. We think that the stability of nanoemulsions is the next step that should be further investigated as mentioned in the previous response. Here, we have proven that the nanoemulsion form demonstrated the enhance efficacy over a short period of time (15 minutes for α-glucosidase enzyme inhibition and 24 h incubation for biofilm formation inhibition). To address these limitations, we have revised the discussion part in Line 500 as follows.
“Moreover, it should be noted that while the nanoemulsified C. nutans extracts demonstrated enhanced efficacy within a short period (15 minutes for α-glucosidase inhibition and 24 hours for biofilm formation inhibition), the stability of these nanoemulsions requires further investigation. As a limitation of the current study, the long-term stability and sustained efficacy of the nanoemulsions have not been explored. Future research should focus on assessing the stability of the nanoemulsions over extended periods to better understand their therapeutic potential in clinical settings.”

Round 2
Reviewer 3 Report
Comments and Suggestions for Authors
The work has been revised according to the suggestions made. I still have a few comments to make:
- Chapter 2: Materials and Methods, should describe how the nanoemulsion particle size analysis was performed.
- Section: References
Literature references should be written in accordance with the journal's requirements: (Author 1, A.B.; Author 2, C.D. Title of the article. Abbreviated Journal Name Year, Volume, page range.). e.g. :
J. Oral Microbiol.( Journal of Oral Microbiology )
Int. J. Health Sci. (Qassim) (International Journal of Health Sciences)
Indian J. Pharmacol. (Indian Journal of Pharmacology )
- Line 146: the % symbol is missing from the formula below.
Percentage Yield = (Final weight of extract/ Initial weight of extract) × 100
- Proper names should be written with a capital letter, e.g.: Span 80, Tween 20.
- Line 262: Please elaborate on the abbreviations: MCT, B-wax
- Line 211: Please provide concentrations of C. nutans extract.
Author Response
The work has been revised according to the suggestions made. I still have a few comments to make:
Response: Many thanks again for the thorough review of our manuscript. We have revised accordingly as follows.
Comment 1
- Chapter 2: Materials and Methods, should describe how the nanoemulsion particle size analysis was performed.
Response 1
Thank you for your comment, we have added more details on the nanoemulsion size analysis in the materials and methods section in Line 274 as follows.
“Therefore, the size of nanoemulsion was investigated by NanoSizer. The nanoemulsion stock solution was diluted in deionized water at a 1:10,000 (v/v) ratio. A 100 µL aliquot of the diluted solution was then transferred to a cuvette for analysis. Particle size distribution was determined using dynamic light scattering (DLS) with a Horiba SZ-100V2 instrument (Horiba, Japan). To ensure accuracy and reproducibility, at least three consecutive measurements were performed. The average particle diameter was calculated from the peak maxima of the DLS size distribution curves, nanoparticle size must not exceed 200 nm.”
Comment 2
- Section: References
Literature references should be written in accordance with the journal's requirements: (Author 1, A.B.; Author 2, C.D. Title of the article. Abbreviated Journal Name Year, Volume, page range.). e.g. :
- Oral Microbiol.( Journal of Oral Microbiology )
Int. J. Health Sci. (Qassim) (International Journal of Health Sciences)
Indian J. Pharmacol. (Indian Journal of Pharmacology )
Response 2
Many thanks for your comment and apologies for the mistakes, we have gone through the references and revised accordingly.
Comment 3
- Line 146: the % symbol is missing from the formula below.
Response 3
Thank you for your comment, we have revised the formula to add % symbol in Line 146 as follows.
Percentage Yield = (Final weight of extract/ Initial weight of extract) × 100%
Comment 4
- Proper names should be written with a capital letter, e.g.: Span 80, Tween 20.
Response 4
Thank you for your comment. We have gone through the manuscript and made sure that the names of chemicals are properly appeared as follows.
Line 263: span 80 has been changed to Span 80.
Line 266: tween 20 has been changed to Tween 20.
Comment 5
- Line 262: Please elaborate on the abbreviations: MCT, B-wax
Response 5
Thank you for your comment. We have gone through the manuscript and made sure that all abbreviations are properly appeared as follows.
Line 263: MCT oil has been changed to Medium-Chain Triglyceride (MCT) oil.
Line 264: B-wax has been changed to bee wax.
Comment 6
- Line 211: Please provide concentrations of C. nutans extract.
Response 6
Thank you for your comment. The concentrations of the C. nutans extract used in each experiment were different ranging from 0-500 mg/mL. The manuscript has been revised to include the concentrations used in each experiment in the materials and methods to make it clearer to the readers as follows.
Line 154:
“A 6 mm cork borer was used to create wells in the agar, into which C. nutans extracts (500 mg/mL) were carefully dispensed”
Line 170:
“The MIC was determined using the broth dilution method, where serial twofold dilutions of the extracts (0-500 mg/mL) were prepared in a 96-well microtiter plate.”
Line 188:
“To assess the anti-biofilm formation and biofilm degradation efficiency of C. nutans extracts, bacterial cultures were prepared at a concentration of 108 cells/mL and inoculated into a 96-well culture plate, with certain wells treated with the extracts at different concentrations shown in Table 2.”
Line 209:
“A reaction mixture was prepared by combining 50 μL of phosphate buffer (pH 6.8), 20 μL of C. nutans extract at various concentrations (0-500 mg/mL), and 10 μL of α-glucosidase enzyme solution (1 U/mL).”
Line 251:
“RAW264.7 macrophages were plated at 1x106 cells/mL in a 96-well plate and treated with or without extracts (95M, 95E, 70E, W) at concentrations of 0.08-2.5 mg/mL for 24 hours followed by stimulation with or without LPS 1 µg/mL for 24 hours.”
